



# The influence of typhoons on atmospheric composition deduced from IAGOS measurements over Taipei

Frank Roux[1], Hannah Clark[2], Kuo-Ying Wang[3], Susanne Rohs[4], Bastien Sauvage[1], Philippe Nédélec[1]

[1] Laboratoire d'Aérologie,Université de Toulouse and Centre National de la Recherche Scientifique, Toulouse, FR-31400 France

[2] IAGOS-AISBL, 98 Rue du Trône, Brussels, B-1050 Belgium

[3] Department of Atmospheric Sciences, National Central University, Taoyuan City, Taiwan 320

[4] Forschungszentrum Juelich GmbH, Institut fuer Energie- und Klimaforschung 8: Troposphaere, DE-52425 Julich, Germany

*Correspondence to:* Frank Roux (frank.roux@aero.obs-mip.fr)

**Abstract :** The research infrastructure IAGOS (In-Service Aircraft for a Global Observing System) equips commercial aircraft with instruments to monitor the composition of the atmosphere during flights around the world. In this article, we use data from two China Airlines aircraft based in Taipei (Taiwan) which provided daily measurements of ozone, carbon monoxide and water vapor throughout the summer of 2016. We present time series from the surface to the upper troposphere, of ozone, carbon monoxide and relative humidity near Taipei, focusing on periods influenced by the passage of typhoons. We examine landing and take–off profiles in the vicinity of tropical cyclones using ERA–5 re–analyses to elucidate the origin of the anomalies in the vertical distribution of these chemical species.

Results indicate a high ozone content in the upper to middle troposphere upstream of the storms. The high ozone mixing ratios are generally correlated with potential vorticity and anti–correlated with relative humidity, suggesting stratospheric origin. These results suggest that tropical cyclones participate in transporting air from the stratosphere to troposphere and that such transport could be a regular feature of typhoons. After the typhoons passed Taiwan, the tropospheric column is filled with substantially lower ozone mixing ratios due to the rapid uplift of marine boundary layer air. At the same time, the relative humidity increases, and carbon monoxide mixing ratios fall. Locally, therefore, the passage of typhoons has a positive effect on air quality at the surface, cleansing the atmosphere and reducing the mixing ratios of pollutants such as CO and O3.

## 1 Introduction

Tropospheric ozone (O3) is an important trace gas from a climate and air–quality perspective. In the upper troposphere, small perturbations to the abundance of tropospheric ozone have relatively large radiative effects (Ramanathan et al., 1987; Forster and Shine, 1997; Riese et al., 2012). Near the surface, ozone has harmful effects on human health and can significantly reduce agricultural yields (e.g. Yue and Unger, 2014). Tropospheric ozone is formed from the action of sunlight on precursors such as carbon monoxide (CO), nitrogen oxides ($NO_x = NO + NO_2$) and volatile organic compounds which are emitted through natural processes and by human activities. There is also a contribution to tropospheric ozone from stratosphere to troposphere transport (Holton et al 1995) in synoptic events along the polar or subtropical jet stream such as tropopause folds and cut–off lows (Stohl et al., 2003; Gettleman et al., 2011), and in relation with deep convection associated with synoptic–scale circulations or mesoscale convective systems (Pan et al., 2014). Lightning activity is another source of ozone through the production of nitrogen oxides which are efficient ozone precursors (Lelieved and Crutzen, 1994). It is important to monitor the abundances and variability of tropospheric ozone and its precursors to better understand the budget of ozone in the troposphere, to understand impacts on surface air quality, and because the long range transport of air pollutants has effects on global atmospheric composition.



The composition of the atmosphere is regularly monitored by IAGOS (In–Service Aircraft for a Global Observing System, a European Research Infrastructure, Petzold et al., 2015; Nédélec et al., 2015) where instruments carried on commercial airlines measure ozone, carbon monoxide and water vapor along with meteorological parameters and cloud particles. Ninety percent of the data are acquired in the upper troposphere-lower stratosphere (UTLS) when the aircraft attain
cruise altitude somewhere between 300 and 180 hPa (9 to 12 km above mean sea level). At these altitudes, the correlations amongst the different trace gases can give insights into various mixing processes near the tropopause (e.g Hoor et al., 2002; Pan et al., 2004; Hegglin et al., 2009; Tilmes et al., 2010). Brioude et al. (2006) used IAGOS observations and Lagrangian analysis to confirm that a layer of enhanced ozone with low carbon monoxide and low relative humidity was due to a tropopause fold associated with a mid–latitude cyclone. More recently, a stratospheric intrusion associated with tropopause
folds along the jet–stream axis of an upper level trough over Europe was captured by IAGOS observations (Akriditis et al., 2018). It led to increased ozone at the surface with negative consequences for air quality.

Tropical cyclones (TCs) – also named Hurricanes in the Atlantic and the eastern North Pacific, and Typhoons in the western North Pacific – are deep convective synoptic–scale systems that can persist for several days. They are associated with significant vertical transport that can make significant perturbations to the structure and chemical composition of the
UTLS. Moreover, strong convection in the eyewall region weakens the tropopause stability and facilitates stratosphere–troposphere exchange with possible ascent of humid and ozone poor air from the lower troposphere and descent of dry and ozone rich air from the lower stratosphere (Romps and Kuang, 2009; Zhan and Wang, 2012). Several authors (Baray et al., 1999; Leclair de Bellevue et al., 2007; Das, 2009; Das et al., 2011 and 2016; Jiang et al., 2015) have emphasized the role of TCs in inducing stratosphere to troposphere exchange accompanied with increases in tropospheric ozone content. However,
the structure and intensity of TCs vary with time and observations of associated ozone perturbations have produced contrasting results. Penn (1965) reported increased values of ozone in the upper troposphere over the eye region of the Hurricane Ginny (1963), but Penn (1966) showed no significant variation in ozone mixing ratio in the lower stratosphere down to the tropopause level above Hurricane Isbell (1964) core. Newell et al. (1996) sampled the chemical structure of Typhoon Mireille (1991) during the Pacific Exploratory Mission –West A campaign in 1991. They showed evidence of
boundary layer air transported to the upper troposphere, but no evidence of ozone of stratospheric origin in the eye region. Likewise, the reduced ozone content measured over and within the inner core region of TC Davina (1999) in southwestern Indian ocean with the stratospheric research aircraft M55 Geophysica suggested a vigorous uplift of ozone–poor oceanic boundary layer air up to 16 km altitude, but no stratospheric intrusion of dry ozone–rich air was detected in the upper troposphere (Cairo et al., 2008). The EP/TOMS (Earth Probe / Total Ozone Mapping Spectrometer) total ozone data for 11
North Atlantic hurricanes (1996–2003) and one western North Pacific typhoon (2001) analyzed by Zou and Wu (2005) revealed that variations of total ozone column (TOC) are closely related to intensity changes of TCs. Intensified deep convection transported ozone–poor air from the oceanic boundary layer in the upper troposphere and caused a decrease of TOC. Likewise, Midya et al. (2012) showed that TCs over the Bay of Bengal and the Arabian Sea for the period 1997–2009 exhibited similar changes with TOC decreasing steadily during the formation and intensification of a cyclone, but increasing
during its dissipation. Lightning activity in tropical cyclones is another potential source of ozone through the associated production of nitrous oxides. Analyzing ozone profiles downstream of Atlantic hurricanes, Jenkins et al. (2015) found increased ozone mixing ratios in the middle and upper troposphere they associated with outflows of lightning induced NOx from intense convective cells in the eyewall and the outer rainbands. Lightning outbreaks in tropical cyclones occur mostly during intensification phases (e.g. Xu et al., 2017), but some weakening storms can also be very active (e.g. DeMaria et al.,
80 2012).

The western North Pacific has some of the most intense cyclonic activity worldwide. On average, six typhoons make landfall annually on the southern China coast (Zhang et al., 2013) where they have been shown to have consequences for the abundance of surface and tropospheric ozone. Major summer pollution episodes in Hong Kong were related to the



presence of tropical storms in the western North Pacific or China Seas (Lee et al., 2002; Huang et al., 2005; Wei et al., 2016), when the low–level northwesterly to westerly winds induced by the storms prior to landfall transported ozone precursors from anthropogenic sources in the urbanized and industrialized Pearl River Delta in Guangdong province, China. Stable weather conditions with calm winds, low humidity, strong solar radiation and high temperatures that prevailed when typhoons were 500 to 1000 km offshore were favorable environment for active photochemical reactions and contributed to the occurrence of the ozone episodes. Likewise, Hung and Lo (2015) showed that increase in surface ozone concentration over southwestern Taiwan two to four days before the passage of typhoons was mainly due to leeward side effects when the cyclonic easterly or northeasterly flow was partially blocked by the Central Mountain Range. The resulting atmospheric subsidence and stable weather conditions reduced vertical mixing and provided a favorable environment for ozone locally produced by photochemical reactions under strong solar radiation to accumulate in the lower troposphere over southwestern Taiwan.

Since 2012, China Airlines has been operating two IAGOS–equipped aircraft from its base in Taipei, Taiwan. Taiwan is situated in one of the most active paths for the TCs that form in the western North Pacific in the Northern Hemisphere and experiences three to four TCs per year (Chen et al., 2015). Here we use the IAGOS dataset over the period 1 July to 31 October 2016 to investigate the influence of typhoons passing near Taiwan on ozone content and tropospheric dynamics. Section 2 gives information on the data and the methods used to analyze them. Section 3 presents the mean profiles of ozone, carbon monoxide and relative humidity from the surface to the upper troposphere from July to October, deduced from IAGOS measurements made during take–off from and landing to Taipei and from ERA–5 re–analyses. Section 4 discusses the vertical profiles of ozone and carbon monoxide from IAGOS data, in relation with PV, relative humidity, and vertical velocity fields from ERA–5 re–analyses. We show examples for three typhoons (typhoons Nepartak on 6–7 July 2016, Nida on 30–31 July 2016 and Megi on 25–26 September 2016) where the IAGOS data allow to identify a clear subsidence of ozone rich air coming from the tropopause region Section 5 puts these results in the context of similar observations over other basins. Section 6 gives the conclusion and some perspectives.

## 2 Data and methods

### 2.1 IAGOS Data

Full details of the IAGOS system and its operation can be found in Petzold et al. (2015) and Nédélec et al. (2015). Ozone is measured using a dual–beam Ultra–Violet – absorption monitor with a response time of 4 seconds and an accuracy estimated at about ±2 parts per billion by volume (ppbv) (Thouret et al., 1998). Carbon monoxide is measured with an infrared analyser with a time resolution of 30 seconds (7.5 km at cruise speed of 900 km h$^{-1}$) and a precision estimated at ±5 ppbv (Nédélec et al., 2003). Test flights showed the stability of the measurements at concentrations above 40 ppbv and a minimum detectable concentration of 10 ppbv.

Water vapor mixing ratio is measured by the IAGOS Capacitive Hygrometer (Neis et al., 2015). IAGOS data provide values of relative humidity with respect to liquid water (RHL). The absolute uncertainty on RHL is estimated to ±5% with a response time of about 1 minute at cruise altitude. The relative humidity (RH) values have been recalculated here using water vapor volume mixing ratio, air temperature and pressure, to account for saturation with respect to liquid water at temperatures warmer than 0°C, to ice at temperatures colder than -20°C, with linear interpolation between these two limits, using the Goff and Gratch (1946) equations which are generally considered as a reference (e.g. Gibbins, 1990).

China Airlines equipped their first aircraft in 2012, becoming the first Asian carrier to join IAGOS. Clark et al., (2015) described the composition of the UTLS over the northern Pacific using data collected at cruise altitude during the first two months of operation. A second China Airlines aircraft was equipped in July 2016, and a third in July 2017. During the





summer of 2016, two aircraft gave up to 5 flights a day from Taipei offering transects in the UTLS and profiles at airports
around the North Western Pacific that were affected by typhoons in 2016.

### 2.2 ERA–5 re–analyses

ERA5 is the fifth generation of the European Centre for Medium–Range Weather Forecast (ECMWF) atmospheric
re–analyses of the global climate (Hersbach and Dee, 2016). It was produced using 4D–Var data assimilation in CY41r2 of
ECMWF's Integrated Forecast System (IFS) coupled to a soil model and an ocean wave model. ERA5 provides hourly
estimates of a large number of atmospheric, land and oceanic climate variables. The data cover the Earth on a 30 km grid and
resolve the atmosphere using 137 hybrid sigma/pressure levels from the surface up to a height of 80 km. ERA-5 re-analyses
offer a more precise description of the atmosphere with improved spatial resolution, better physics, advanced modelling and
data assimilation compared to the previous ERA-Interim re-analyses. (Dee et al., 2011).

### 2.3 The use of potential vorticity

Potential vorticity (*PV*) is defined as the scalar product of the absolute vorticity vector $\boldsymbol{\zeta_a}$ and the gradient of
potential temperature θ, divided by the air density ρ:

$$PV = \frac{\boldsymbol{\zeta_a} . \boldsymbol{\nabla}\theta}{\rho} = \frac{1}{\rho}(2\boldsymbol{\Omega} + \boldsymbol{\nabla} \times \boldsymbol{v}) . \boldsymbol{\nabla}\theta$$

(1)

where $\boldsymbol{\Omega}$ is the angular velocity vector of the Earth's rotation, $\boldsymbol{v}$ the three–dimensional velocity relative to the
rotating earth, $\boldsymbol{\nabla}$ is the three–dimensional vector differential operator. PV is conventionally expressed in PV Units (1 PVU
$= 10^{-6}$ K m$^2$ kg$^{-1}$ s$^{-1}$). The PV tendency equation can be written as:

$$\frac{\partial PV}{\partial t} + (\boldsymbol{v} . \boldsymbol{\nabla}) PV = \frac{1}{\rho}(\boldsymbol{\nabla} \times \boldsymbol{v}) . \boldsymbol{\nabla}\theta + \boldsymbol{\nabla}\dot{\theta}$$

(2)

where $\boldsymbol{F}$ is the friction or dissipation force and $\dot{\theta}$ is the diabatic heating rate.

The troposphere and the stratosphere have very different properties, in terms of relative humidity, PV, and chemical
species like ozone. The air is moister in the troposphere than in the stratosphere, whereas PV values and ozone content are
higher in the stratosphere. Since potential vorticity is conserved in adiabatic and frictionless flow, an intrusion of
stratospheric air into the upper troposphere can be followed in space and time by considering anomalies of PV and RH. In
tropical cyclones, the rate of latent heating peaks in the mid–troposphere, so the rate of PV creation is positive below this
maximum and negative above it, leading to a vertically oriented dipole. As the heating induced radial–vertical circulation
develops, the positive (cyclonic) PV values in the lower troposphere are carried aloft, while the negative (anti-cyclonic) ones
in the upper troposphere are dragged by the upper level outflow to larger radii. Therefore, a mature vortex is characterized by
a positive PV anomaly extending through much of the troposphere, with negative values in a wider but shallower layer
below the tropopause.

### 3 Vertical profiles of tropospheric ozone, carbon monoxide and relative humidity at Taipei

Figure 1 shows the mean vertical profiles of ozone, carbon monoxide, and relative humidity calculated from over
250 profiles collected by IAGOS–equipped aircraft for 1 July to 31 October 2012–2015 during take–off from and landing to
Taoyuan International Airport (International Air Transport Association code : TPE ; 25.076°N, 121.224°E) located about 40
km west of Taipei,. Ozone mixing ratios in the free troposphere, at altitudes above 2 km, are typically 40 to 50 ppbv. The





surface layer shows a larger variability with mixing ratios ranging from 40 to 160 ppbv. The mean ozone content in 2016 was slightly less than the mean 2012–2015 values at altitudes below 4 km and greater at altitudes above 6 km. The mean
mixing ratio of carbon monoxide above 1 km is around 110 ppbv, increasing sharply below to ≈250 ppbv at the surface, due to local pollution from the urban and industrial environment. In 2016, the mixing ratios of CO above 2 km were lower than those in 2012–2015. Near the surface, mixing ratios reach 320 ppbv which is slightly higher than in 2012-2015. The mean relative humidity is typically 80% at the surface, and it decreases almost linearly to ≈30% at 10  km altitude and above. In 2016, the mean relative humidity below 4 km was lower than in 2012-2015. At altitudes above 4 km, it was greater than in
2012-2015. A possible explanation could be that 2016 had a more frequent occurrence of deep convective clouds which transported the low–level humidity to the upper troposphere.

More precisely, the evolution of the vertical profiles of relative humidity, ozone and carbon monoxide from 1 July to 31 October 2016 reveal distinct periods (Fig. 2). Beforehand, comparisons with vertical profiles of mean relative humidity [using the same Goff and Gratch (1946) equations and temperature thresholds], potential vorticity, and vertical velocity
derived from ERA–5 re–analyses within 300 km from TPE for the same period (Fig. 3) allow us to place the IAGOS measurements in their meteorological context. Although there are some differences between both RH values, the overall alternation of dry (RH < 50%) and wet (RH > 50%) episodes is remarkably similar.

Wet periods in Figs. 2a and 3a were associated with mean upward motion (Fig. 3c) and relatively low (<50 ppbv) ozone content (Fig. 2b), corresponding to the upward transport of relatively clean air from the nearby oceanic boundary layer
associated with the occurrence of convective systems. Among these events, typhoons that passed close to TPE are indicated with black lines in Figs. 2 and 3: Nepartak on 7 July 00 UTC, Nida on 31 July 12 UTC, Meranti on 13 September 00 UTC, Malakas on 16 September 12 UTC, Megi on 26 September 00 UTC, Sarika on 16 October 00 UTC, Haima on 19 October 12 UTC. Four of them, Nepartak, Meranti, Malakas, and Megi, were close enough to TPE to be also identified with positive PV values in the mid– to upper troposphere  (Fig. 3b). Other moist periods on 18–20 July, 10–12 August, 5–10 September, 8–10
October, were associated with more or less organized convective systems embedded in larger scale ensemble spanning over the South and/or the East China Seas.

Except for a wet episode on 10–12 August, the month of August was generally characterized by dry RH (<50%), relatively large O3 (>50 ppbv) and CO (>100 ppbv) content above 3 km altitude over TPE ( Fig. 2 ). ERA–5 data shows that, during this period, mean vertical motions were predominantly downward (Fig. 3c) and that potential vorticity reached
unusually high values (>1 PVU, Fig. 3b) down to 6 km altitude on 9, 19 and 25 August. Himawari–8 satellite images in the water vapor / mid–infrared (6 – 7.5 μm wavelengths) channels 8, 9, 10 (not shown) indicate that in August 2016 Taiwan was only marginally affected by perturbed weather as dry conditions prevailed over most of the East China Sea. It is therefore reasonable to consider that the relatively high ozone content in August 2016 originated from the stratosphere in mostly anti–cyclonic and subsiding conditions in the local upper troposphere. It is worth noting that the highest mixing ratios of ozone
(up to 100 ppbv, Fig. 2b) and carbon monoxide (up to 350 ppbv, Fig. 2c) below 2 km in altitude were observed on 29–31 August, just before a moist and cloudy zone moved westward and occupied a large zone from northern Vietnam to Japan.

From Figs. 2 and 3, it is difficult to clearly identify the influence of typhoons passing close to TPE on the physical and chemical characteristics of the troposphere. The following section shows more detailed analyses of the IAGOS and ERA–5 data for three typhoons in 2016.

**4 Physical and chemical impact of three typhoons**

The 2016 typhoon season in the western North Pacific was an average one, with a total of 26 named storms and 13 typhoons (RSMC Tokyo, 2017; JTWC Guam, 2017). Seven of them (typhoon Nepartak on 06–07 July, severe tropical storm Nida on 30–31 July, typhoon Meranti on 11–13 September, typhoon Malakas on 16 September, typhoon Megi on 25–26





September, typhoon Sarika on 15–16 October, typhoon Haima on 19–20 October) went close enough to Taiwan for vertical
profiles of ozone, carbon monoxide and relative humidity to be obtained at less than 1000 km from their center by IAGOS
aircraft taking off from or landing to TPE. The most favorable typhoon trajectories and detailed observations occurred for
typhoons Nepartak, Nida and Megi (thick black lines in Figs. 2 and 3). Below, we combine IAGOS observations for these
three storms with images taken by the Japanese Himawari–8 geostationary satellite, and relative humidity, potential vorticity
and vertical velocity fields from ERA–5 re–analyses, in order to place them in meteorological context, and to detail the
influence of typhoons on the physical and chemical characteristics of the local troposphere. The fine resolution (≈30 km
horizontally) of ERA-5 re-analyses provide detailed information that were not available in previous studies.

**4.1 Typhoon Nepartak on 06–07 July 2016**

Starting as a low pressure area south of Guam (13.50° N , 144.80° E) on 30 June, then a tropical depression on 2
July and then a tropical storm on 3 July, Nepartak became a typhoon on 4 July. It reached its peak intensity on 6 July 12
UTC as a Category 5 equivalent super–typhoon with a central pressure at 900 hPa and 10 min averaged maximum winds at
205 km h$^{-1}$ (≈55 m s$^{-1}$) (Fig. 4). At that time, its center was located about 700 km to the southeast of Taiwan. Nepartak started
to weaken on 7 July afternoon when its circulation began to interact with the topography of Taiwan. It crossed Taiwan, then
emerged into the Taiwan Strait as a weaker Category 1 typhoon on 8 July. Nepartak made final landfall one day later in
China's Fujian Province.
Relevant IAGOS observations were collected during six selected takeoffs and landings at TPE from 6 July 0457
UTC until 7 July 1118 UTC, numbered from NEP-1 to NEP-6 (in Fig. 5). To put these quasi--vertical profiles in the
meteorological environment of typhoon Nepartak, we positioned them in a reference frame moving with the storm at 7 m s$^{-1}$
from 295° with its origin at 12.18°N, 125.10° E on 6 July 00 UTC. In this reference frame, profile NEP-1 is the furthest from
the centre of Nepartak  and NEP-6 is the closest as the airport gets closer with time to the centre of the typhoon. Figure 6
shows the result from the combination of seven six-hourly ERA–5 re–analyses from 6 July 00 UTC till 7 July 12 UTC in this
reference frame moving with Nepartak. The white dotted line, referred to as "S axis" in Fig. 6a, represents thepseudo-path of
TPE airport with respect to Nepartak (rather than the storm track of Nepartak relative to the airport). Figure 6a shows the
horizontal distribution of relative integrated humidity (RIH), deduced from actual and saturated mixing ratios over altitudes
between 4 and 10 km. We can see the moist central and eastern parts of Nepartak with RIH ≥ 80 %, and a large region of dry
air with RIH ≤ 40% to the northwest. There is a very good correlation between moist and dry zones from ERA-5 (Fig. 6a),
and the bright and dark regions from Himawari-8 Channel 8 / Water Vapor images (Fig. 4).  The mean potential vorticity
(MPV) field between 8 and 12 km altitude (Fig. 6b) reveals large positive values (>2 PVU) in the upper core region of
Nepartak, and a south–north oriented band of weaker positive values (>0.5 PVU) at the eastern limit of the dry zone. This
region was associated with mean downward motion at altitudes between 6 and 10 km, shown by the black line in Fig. 6b.
The descent rate was greater than -1 cm s$^{-1}$ which is significantly larger than the typical value for tropical clear–sky regions
(Gettelman et al., 2004; Das et al., 2016).
In Figures 6c and d, we present cross-sections of the relative humidity and PV averaged over ±500 km on either side
of the pseudo-path of TPE in the frame moving with the typhoon ("S axis" in Fig. 6a). Comparison of Fig. 6c and d reveals
that the dry zone was related to a region of air with MPV values larger than 1 PVU, originating from the tropopause region
near 15 km altitude. Though such values are less than the 2-PVU threshold commonly used to characterize stratospheric air,
they are significantly higher than those observed in the troposphere, except in the vicinity of strong cyclonic perturbations
such as typhoons. These high MPV values in the upper troposphere were also associated with downward motion (< -1 cm s$^{-1}$)
and dry air (RIH < 40 %) down to 3 km altitude (Fig. 6c). This feature is distinct from the high-MPV and moist-RIH (>60%)
values associated with the core region of typhoon Nepartak. Above 10 km, negative MPV values with nearly saturated RIH



below the tropopause, at 15 km altitude, resulted from the divergent and anti–cyclonic moist outflow from Nepartak. It should be noted that the dry RIH and high MPV zone seen in Figs. 4 and 6a and b was not stationary over the Taiwan Strait and mainland China, but rather it moved with Nepartak on 6–7 July staying 500 to 1500 km to the northwest with limited deformation.

From the correlations amongst the different chemical species measured by IAGOS, we can identify different air
masses which are present at different altitudes within the successive profiles. In the UTLS over the north Pacific, IAGOS aircraft observed air masses with high humidity and low concentration of pollutants CO (<100ppbv) and ozone (20-40ppbv) indicative of air from the marine boundary layer probably lifted aloft by deep convection (Clark et al. 2015). We focus here on identifying the chemical characteristics of the dry layers which are influenced by subsiding motion a few hundred kilometers ahead of typhoon.

The first four vertical profiles derived from IAGOS measurements (NEP-1 to NEP-4 in Fig. 5) reveal that the dry layer above 5 km altitude was associated with a large ozone content, up to 100 ppbv. The mixing ratios of carbon monoxide seen at these altitudes were significantly lower than the values of 200 to 250 ppbv observed near the surface. Hence, the dry, $O_3$ rich and CO poor zone aloft probably did not result from the upward transport of polluted boundary layer air, nor was the ozone likely to be the result of lightning activity as this would be a high ozone and moist air combination (Jenkins et al.
2015). These four profiles lie within the region of subsiding motion as shown by the black contour on Fig. 6d and in addition, intercept the tongue of higher PV air originating from the region above 15km (Fig. 6d) suggesting that the high ozone is a result of stratosphere to troposphere inflow ahead of the approaching typhoon.

Figure 6d shows that profiles NEP-5 and NEP-6 were obtained in the more humid region close to Nepartak, except for the southern tip of the dry and ozone rich zone layer intercepted by profile NEP-6 at 6 km altitude which may be the
remnants of a previous mixing or intrusion event. At altitudes above 6 km, these profiles differ from the first four as the ozone mixing ratios have dropped by 70 ppbv to around 20 ppbv, and the relative humidity has increased from 10 to 80%. This humid, ozone and CO poor layer resulted probably from upward transport of clean humid air from the oceanic boundary layer by the typhoon. At altitudes below 2 km, profiles NEP-4, 5, 6 which are much closer to the typhoon reveal that, when the center of Nepartak was at less than 500 km from Taipei, the characteristics in the lower atmosphere below 2 km altitude
changed dramatically with a strong decrease of CO and increase in relative humidity consistent with inflow of cleaner and moist oceanic boundary layer air feeding the typhoon at the lower levels.

Below we present a further two typhoons, in order to determine how common these features are.

**4.2 Severe tropical storm Nida on 30–31 July 2016**

Formed as a tropical depression over the Philippine Sea during the night of 29 to 30 July, Nida followed a
northwesterly track and intensified in the afternoon of 30 July. After developing into a severe tropical storm the next morning, it passed close to the north of Luzon (Philippines) on 31 July (Fig. 7). When it entered the South China Sea, Nida further intensified into a severe tropical storm and reached its peak intensity on 31 July 09 UTC with maximum 10 min averaged winds at 110 km h$^{-1}$ (≈30 m s$^{-1}$) and minimum central pressure at 975 hPa. At that time, the storm center was about 400 km to the south–southeast of Taiwan. Nida made landfall near Dapeng Peninsula, east of Hong Kong, in the evening of 1
August, before weakening as it moved further inland and dissipating on 2 August.

IAGOS observations were obtained during six flights from TPE from 30 July 0955 UTC until 31 July 2140 UTC (Fig. 8). As in Fig. 6 above for the typhoon Nepartak, the profiles for Nida were put in their meteorological context derived from the combination of six–hourly ERA–5 re–analyses from 30 July 06 UTC until 01 August 00 UTC in the reference frame moving with Nida, at 6 m s$^{-1}$ towards 305° with its origin at 7.4°N, 123°8 E on 30 July 00 UTC. The white dashed line
(referred to as S axis) in Fig. 9a again shows the pseudo-track of TPE airport in the reference frame of Nida . A comparison





between Figs. 6a and 9a reveals that Nida passed at a larger distance from TPE than Nepartak. The relative integrated humidity between 4 and 10 km altitude (Fig. 9a) shows the moist central and eastern parts of Nida with RIH ≥ 80 %, and as with Nepartak, a large region of dry air (RIH ≤ 40%) to the northwest. The mean potential vorticity between 8 and 12 km altitude (Fig. 9b) also reveals relatively large positive values (> 1 PVU) in the upper core region of Nida, and a wide band

with weaker positive values (> 0.5 PVU) in the dry region to the northwest. Mean downward motions between 6 and 10 km stronger than -1 cm s$^{-1}$ are also observed in this dry zone.

Mean cross–sections within ± 500 km along the vertical domain of the IAGOS measurements show that the dry zone was related to an intrusion of high MPV (>1 PVU) air originating from the lower stratosphere into the troposphere. In both the Nepartak and Nida cases, there is a tongue of high MPV air downstream of the typhoon. These high MPV values in

the upper troposphere at S > 1000 km are associated with downward motions (< - 1 cm s$^{-1}$) between 10 and 15 km altitude, and dry air (RIH < 40 %) down to 4 km (Fig. 9c). The low RH and relatively high MPV zone moved with Nida on 30-31 July while staying 500 to 1500 km to the northwest with limited deformation. The high-MPV zone at 500 < S < 1000 km is rather associated with the production of positive PV through latent heating and its vertical transport in the core region of Nida.

The vertical profiles NID-1 to NID-6 from the IAGOS aircraft measurements before the arrival of Nida ( Fig. 8) show again that the dry layer aloft was associated with a large ozone content (up to 100 ppbv). The bottom of the ozone rich layer was between 3 and 4 km altitude, and its top was higher than 11 km, the flight level of IAGOS aircraft. As with Nepartak, this indicates that the dry, O3- and relatively CO-poor zone above 3 to 4 km altitude did not result from the upward transport of polluted boundary layer air, but more probably from stratosphere to troposphere inflow revealed by

potential vorticity, relative humidity and vertical velocity fields (Fig. 9c and d).

Profile NID-6, in the more humid region to the northeast of Nida (Fig. 9c), shows a much lower ozone content (<30 ppbv) and more humid air, along with low CO throughout the free troposphere up to 8 km altitude. This profile is similar to profile NEP-5 (and NEP-6 except near 6km) in Fig. 5 from the area near to Nepartak. Profile NID-6 appears to encounter the tongue of higher MPV air, (> 0.5 PVU, Fig. 9d)  indicated by the ERA-5 analyses, but IAGOS instruments have not detected

an increase in ozone. The correlation with high relative humidity (>80%) suggests that this high-PV air probably originated from the cyclonic circulation associated with Nida.

### 4.3 Typhoon Megi on 25–26 September 2016

Originating from a tropical disturbance northeast of Pohnpei (6.88°N, 158.23°E) on 19 September, Megi was identified as a tropical storm on 23 September, and upgraded to a typhoon on 24 September. After a pause in its development

on 25 September, Megi strengthened again on 26 September afternoon (Fig. 10). It reached its maximum intensity on 27 September 00 UTC with 10 min averaged winds up to 155 km h$^{-1}$ (≈45 m s$^{-1}$) and central pressure at 945 hPa, before it made landfall on the eastern coast of Taiwan at 06 UTC. Weakened by the interaction with the Central Mountain Range, Megi emerged into the Taiwan Strait in the early afternoon. It made final landfall over mainland China near Xiamen, Fujian province, in the evening of 27 September.

We use the IAGOS observations at TPE from 25 September 1412 UTC until 26 September 1604 UTC (Fig. 11). The profiles were put in their meteorological context derived from the combination of six–hourly ERA–5 re–analyses from 25 September 12 UTC till 26 September 18 UTC in the reference frame moving with Megi, at 5.5 m s$^{-1}$ towards 300° with its origin at 12.76°N, 124.5° E on 6 July 00 UTC. Relative integrated humidity between 4 and 10 km altitude (Fig. 12a) shows the central and eastern parts of Megi with RIH ≥ 80 %, and a large region of dry air (RIH ≤ 40%) to the north. The MPV

field (Fig. 12b) reveals large positive values (>2 PVU) in the upper core region of Megi, and a band of weaker positive





values (>0.5 PVU) in the dry zone to the west of Megi where mean downward motions < -1 cm s⁻¹ between 6 and 10 km are also observed. .

Mean cross–sections within ±500 km along the vertical domain where IAGOS measurements were made (see S axis in Fig. 12c and d) show that the dry zone was related to an intrusion of air with relatively high potential vorticity (>1 PVU)

from the lower stratosphere into the troposphere (Fig. 11d). These MPV values in the upper troposphere were associated with downward motion (< -1 cm s⁻¹) down to 5 km altitude and dry air (RIH < 40 %) down to 2 km (Fig. 12c). The low RH and relatively high MPV zone moved without much deformation while staying 500 to 1500 km to the north of Megi on 26–27 September.

The vertical profiles MEG-1 to MEG-6 deduced from the IAGOS aircraft measurements (Fig. 11) reveal that the dry

layer aloft was associated with a relatively large ozone content (up to 80 ppbv), though lower than the 100 ppbv seen for Nida and Nepartak. The bottom of this layer was at about 2 km altitude for the first profile MEG-1, and it increased thereafter up to 5–6 km, whereas its top remained at a nearly constant altitude of 9 km. Similar to profiles NEP-5, NEP-6, and NID-5, the MEG-6 profile, in the more humid region close to Megi, has a much lower ozone content (<30 ppbv) and more humid air from the surface up to 9 km altitude, except for a small spike of ozone and dry air at 5.5 km altitude which

represents the southern limit of the dry and ozone rich layer (Fig. 12c). Again, the CO content in this layer is significantly weaker than the values up to 150–200 ppbv observed near the surface which suggests that the dry, O3 rich and CO poor zone did not result from the upward transport of polluted boundary layer air, but more probably from the stratosphere to troposphere inflow revealed by potential vorticity, relative humidity and vertical velocity fields (Fig. 12c and d).

## 5 Discussion

The structures of relative humidity, vertical velocity and potential vorticity were also deduced from ERA-5 data for typhoons Meranti, Malakas, Sarika and Haima (not shown) and were broadly similar to those observed for Nepartak, Nida and Megi. The less favorable coverage by IAGOS aircraft for these typhoons which passed at a larger distance from Taiwan did not permit such detailed comparisons of the O3 and CO profiles. Generally, a tongue of high PV air and ozone richer air is observed ahead of the typhoons. Dry zones (RH < 40%), which could also be identified on Himawari-8 images in the

water vapor channels and were associated with downward motion (< -1 cm s⁻¹), were systematically found 500 to 1000 km ahead or to the northwest of the storms when they approach Taiwan over the Philippine Sea, the Luzon Strait and the South China Sea. The associated potential vorticity signature was more variable with a very strong signal (>1 PVU down to a, altitude of 7 km ) for Malakas on 16 September 2016, a significant signature (>0.5 PVU down to an altitude of 8 km) for Sarika on 16-17 October 2016, and weaker signatures (0-0.5 PVU above 10 km altitude) for Meranti on 11-13 September

2016 and Haima on 19-20 October 2016. These dry subsiding zones associated with the seven typhoons that passed close to Taiwan in 2016 were not directly related to the mid-latitude westerly circulation, since – when present – the upper-level jet (>40 m s⁻¹) was always located between 35 and 40°N over northern China, Korea or Japan, about 1500 km north to northeast of Taiwan. In contrast, closer to the centre, air is reduced in ozone, and more humid consistent with arrival of marine boundary layer air. This is also a repeated feature of the typhoons.

Our results on the influence of tropical cyclones on tropospheric ozone near Taiwan and southern China are comparable with observations over the southwestern and northern Indian Ocean. Baray et al. (1999) attributed high values of ozone (>50 ppbv between 500 and 150 hPa) with low relative humidity (<30% above 500 hPa level) revealed by a radiosonde launched from Saint–Denis, La Réunion (20.88°S, 55.45°E) on 6 April 1995 to subsidence in the upper troposphere west of Tropical Cyclone Marlene whose center on 6 April 1995 was at its nearest to La Réunion, about 1000

km to the east. The dry and ozone rich layer in the upper troposphere corresponded to a relative maximum of PV in the analyses by two general circulation models with horizontal resolution of 2.5° (National Center for Environmental Predictions





/ National Center for Atmospheric Research) and 1.125° (ECMWF). However these resolutions were too coarse to allow explicit determination of the origin of this PV maximum. A complementary numerical simulation using the MésoNH model (Lafore et al., 1998) at higher resolution (45 km) by Leclair de Bellevue et al. (2007) showed that, on 6 April 1995, the
island of La Réunion was under a stratospheric PV filament (>1 PVU) into the troposphere, crossing the isentropes down to the 350 K level (≈200 hPa). This feature was probably related to upper tropospheric zones of divergence and convergence organized as rings from the center to the periphery of the cyclone.

Das (2009) reported evidence of stratospheric intrusion into troposphere, identified as enhanced signal–to–noise ratio of thickness ≈1 km between 13 and 16 km altitude with the mesosphere–stratosphere–troposphere (MST) radar at
Gadanki (13.46°N, 79.18°E; Andhra Pradesh, India) on 16 October 2001. The same day, cyclonic storm BoB–01, with a Dvorak rating of 2.5, made landfall near Nellore, about 150 km northeast of Gadanki. Das et al. (2011) investigated this event with a numerical simulation with the Advanced Research Weather Research and Forecast (WRF–ARW, Skamarock et al., 2005) model at a horizontal resolution of 27 km. Analysis of vertical velocity and PV revealed that the stratospheric intrusion occurred in the periphery of the cyclone center. The horizontal and vertical scales of the intrusion were 200–250
km and 5–6 km, respectively, with a width of about 50 km.

More recently, Das et al. (2016) analyzed series of ozonesondes launched from Trivandrum (8.51°N, 76.96°E; Kerala, India) during two cyclone events from the Bay of Bengal : Cyclonic (tropical) storm Nilam from 30 October to 7 November 2012, and very severe cyclonic storm (equivalent to a Category 1 hurricane) Phailin from 11 to 15 October 2013. Both events were associated with an increase in the upper tropospheric (10–16 km) ozone which propagated downward at a
rate of about -1 cm s$^{-1}$. WRF–ARW simulations revealed that these ozone rich zones were associated with enhanced PV values (0.5 to 1.5 PVU) overlapping the downdraft regions and extending vertically down from the stratosphere to the lower troposphere. The simultaneous presence of dry air from the tropopause level down to 4 km altitude was an additional indication of descent of stratospheric air into the troposphere, induced by the tropical cyclones.

Jiang et al. (2015) reported large increases in surface ozone between Xiamen and Quanzhou from 12 to 14 June
2014, before Typhoon Hagibis made landfall over the southeastern coast of China early on 15 June. Weak easterly winds carrying clear air from the ocean, low nitrogen oxides concentrations, and negative correlation between ozone and carbon monoxide suggested that the surface ozone peak was unlikely caused by horizontal advection from anthropogenic sources. Vertical velocities from NCEP analyses and dry surface air provided indications of strong downward transport of ozone in a subsiding branch about 400 km to the northeast of Typhoon Hagibis center, over the Xiamen – Quanzhou region.

These studies showed that a strong cyclone–driven intrusion of stratospheric air can transport ozone down to the low troposphere and the surface, and lead to noticeable anomalies in its concentrations. It is also possible to establish a parallel between these observations and the report by Pan et al. (2014) of ozone rich stratospheric air wrapping around both leading and trailing edges of a mesoscale convective system and descending down to an altitude of 8 km, about 4 km below the local tropopause level.

## 395 6 Conclusion and perspectives

Fig. 13 shows a schematical cross–section of a model typhoon (or tropical cyclone) and the different ways it can modify the chemical characteristics of the troposphere. The results presented here, and those by some of the authors quoted above, relate mostly to one specific process: upper tropospheric – lower stratospheric intrusion identified by correlations between high ozone and low carbon monoxide contents, downward motions, dry relative humidity, and relatively high PV
values. Although it has been documented on several occasions, it is not clear whether such phenomenon which occurs 500 to 1000 km from the centre of the storm is a frequent feature associated with tropical cyclones worldwide.





We have also seen that the upward transport of clean humid air from the oceanic boundary layer decreases the tropospheric ozone content. Other processes schematically represented in Fig. 13 can also influence the chemical characteristics of the troposphere in the vicinity of tropical cyclones.. Several authors have shown that ozone and carbon 405monoxide would increase when the inflow is at least partially from polluted continental areas. In the cases that we have presented, the pollution generally remained in the boundary layer below 1km. Subsidence above the eye could also carry stratospheric ozone into the upper troposphere, but the concerned region has a relatively limited area and this process might not be very efficient at storm scale. Commercial flights carrying the IAGOS instruments would never fly in such a hazardous region. Lightning associated with intense convection in the eyewall and the external rainbands produce nitrogen oxides and 410ozone which is then advected horizontally and vertically. However, the relationship between tropical cyclones and lightning activity is complex, and it is not certain that this mechanism represents a major contribution to the ozone budget for the majority of storms.

It is therefore necessary to continue the sampling of tropospheric ozone (and carbon monoxide) profiles in the vicinity of tropical cyclones to draw statistically and climatologically reliable figures. The fact that IAGOS aircraft from 415China Airlines operate from TPE airport is an excellent opportunity, considering the frequent occurrence of typhoons in the vicinity of Taiwan. IAGOS offers additional instrumentation to measure NOx which is available on one aircraft so far (Berkes et al 2018). In the future it is hoped that NOx instruments on China airlines could help to determine the amount of lightning NOx produced in the vicinity of typhoons. Of course, it would also be necessary to obtain similar information for storms over other basins to examine whether the observed characteristics relate to the cyclone structure and evolution, or to 420some specificity of their environment. Additional information might also come from numerical simulations with high resolution models explicitly representing convective dynamics, microphysics, electrification, and lightning (e.g. Fierro et al., 2011; Xu et al., 2014; Barthe et al., 2016), as well as the associated production and transport of nitrogen oxides and ozone. It would also be necessary to represent correctly the troposphere – stratosphere interactions associated with tropical cyclones at local and meso– scales (e.g. Dauhut et al., 2018). Such results would help to quantify more precisely the global contribution 425of tropical cyclones to the chemistry budget of the troposphere.

**7 Code availability**

The MATLAB scripts used to analyze the ERA-5 data (Net-CDF files formatted by CLIMSERV) are available at https://mycore.core-cloud.net/index.php/s/vi2SmsVJNVnM4w0)
- "coupe_zt_era5.txt" and "coupe_zt_era5.m" were used to obtain Fig. 3
430The MATLAB scripts used to analyze the IAGOS data (Net-CDF files formatted by IAGOS Data Portal) are available at https://mycore.core-cloud.net/index.php/s/JWmep21ghplwUG1
- "juloct.m" was used to obtain Fig. 2
- "tra_rho3co.txt" and "tra_rho3co.m" were used to obtain Figs. 6, 9 and 12

The MATLAB scripts used to simultaneously analyze ERA-5 and IAGOS data (in Net-CFDF formats) are available at 435https://mycore.core-cloud.net/index.php/s/dscfB8XUqLaw3Pv
- "rhpvw_era5_sv_meanbox.txt" and "rhpvw_era5_sv_meanbox.m" were used to obtain Figs. 5, 8, 11

**8 Data availability**

The ERA-5 data have been obtained through CLIMSERV Data Portal at http://climserv.ipsl.polytechnique.fr/
The IAGOS data have been obtained through IAGOS Data Portal at https://www.iagos-data.fr/





## 9 Authors contributions

FR, HC, KYW, SR, BS and PN conceived the project, devised the main conceptual ideas, and reviewed the literature. FR designed and performed the numerical calculations. HC and FR analyzed the results and wrote the manuscript. FR, HC, KYW, SR, BS and PN conducted critical review.

## 445 10 Competing interests

The authors declare that they have no conflict of interest.

## 11 Acknowledgements

IAGOS is funded by the European Union projects IAGOS–DS and IAGOS–ERI. The IAGOS database is supported in France by AERIS (Centre National d'Etudes Spatiales and Institut National des Sciences de l'Univers / Centre National de 450 la Recherche Scientifique). We acknowledge the strong support of the European Commission, Airbus and the airlines (Lufthansa, Air France, Austrian Airlines, Air Namibia, Cathay Pacific, Iberia and China Airlines so far) that carry the IAGOS equipment.

ERA–5 re–analyses data were obtained from CLIMSERV (http://climserv.ipsl.polytechnique.fr/), a French open access service developed by Laboratoire de Météorologie Dynamique (Paris) and maintained by Institut Pierre Simon 455 Laplace, for studies related to climate, atmospheric processes, technical and scientific analyses of Earth observing satellite data.

Himawari–8 geostationary satellite images in Figs. 4, 7 and 10 were downloaded and are reproduced with permission from NERC Satellite Receiving Station, University of Dundee, Scotland (http://www.sat.dundee.ac.uk/).

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

**13 Figure Captions**

**Figure 1 :** Mean profiles of **(a)** ozone, **(b)** carbon monoxide, **(c)** relative humidity from IAGOS measurement during take–off from and landing to TPE over the time period 1 July to 31 October 2012 to 2015 (in black) along with the standard deviation indicated by the shaded area, and for 2016 (in red). The total number of days with data is indicated.

**Figure 2 :** Vertical profiles derived from IAGOS measurements during take-off from and landing to TPE (Taoyuan International Airport ; 25.076°N, 121.224°E ) from 1 July till 31 October 2016 for **(a)** relative humidity in %, **(b)** ozone mixing ratio in parts per billion by volume – ppbv, **(c)** carbon monoxide in ppbv. Vertical black lines labeled NEP, NID, MER, MAL, MEG, SAR, HAI represent the closest approaches of typhoons Nepartak, Nida, Meranti, Malakas, Megi, Sarika and Haima, respectively.

**Figure 3 :** Vertical profiles derived from ERA-5 re-analyses within 300 km from TPE from 1 July till 31 October 2016 for **(a)** mean relative humidity in %, **(b)** mean potential vorticity in potential vorticity unit – 1 PVU = $10^{-6}$ $m^2$ s-1 K $kg^{-1}$, **(c)** mean vertical velocity in cm $s^{-1}$. Vertical black lines labeled NEP, NID, MER, MAL, MEG, SAR, HAI represent the closest approaches of typhoons Nepartak, Nida, Meranti, Malakas, Megi, Sarika and Haima, respectively.

**Figure 4 :** Himawari-8 Water Vapor / Channel 8 (6.06 - 6.43 μm) image on 6 July 2016 18 UTC showing typhoon Nepartak centered about 550 km southwest of Taiwan. Orange dot labelled TPE indicates the location of the Taiwan Taoyuan International airport. The yellow contour denotes the 2000 km × 2000 km region where ERA-5 re-analyses data were composited (X and Y axes are shown). The storm motion (7 m $s^{-1}$, 295°) is indicated.

**Figure 5 :** Vertical profiles of relative humidity RH (green curves, in %), ozone O3 (blue curves, in parts per billion by 625 volume – ppbv) and carbon monoxide CO (red curves, in ppbv) from the six takeoffs from and landings to TPE of IAGOS aircraft on 6-7 July shown in Fig.5. The dashed line is an arbitrary limit separating dry ( RH <40 %) and moist (RH >40 %), O3-poor (<40 ppbv) and O3-rich (>40 ppbv), CO-poor (<100 ppbv) and CO-rich (>100 ppbv) air masses. Thick orange segments indicate regions with dry and O3-rich air.

**Figure 6 :** Composites of ERA-5 re-analyses data from 6 July 00 UTC till 7 July 12 UTC in the domain shown in Fig. 4. 630 Above are horizontal distributions of **(a)** relative integrated humidity (RIH) between 4 and 10 km altitude (in %), and **(b)** mean potential vorticity (MPV) between 8 and 12 km altitude in PVU (= $10^{-6}$ $m^2$ $s^{-1}$ K $kg^{-1}$). The north direction is indicated in **(a)**. The dotted white line (S axis) represents the pseudo-trajectory of TPE airport in the frame moving with Typhoon Nepartak (successive times are indicated). Below are vertical cross-sections of **(c)** RIH in %, and **(d)** MPV in PVU, averaged across ± 500 km along this oblique S line. The solid black contours 635 encompass mean downward velocities < -1 cm $s^{-1}$ between 6 and 10 km altitude. The purple lines 1 to 6 indicate the tracks of IAGOS aircraft during selected takeoffs and landings on 6-7 July.

**Figure 7 :** As in Fig. 4, except for severe tropical storm Nida on 31 July 2016 00 UTC, and storm motion (6 m $s^{-1}$, 305°).

**Figure 8 :** As in Fig. 5, except for the six takeoffs from and landings to TPE of IAGOS aircraft on 30-31 July.

**Figure 9 :** As in Fig. 6, except for the period 30 July 06 UTC - 01 August 00 UTC, and 6 tracks of IAGOS aircraft during 640 takeoffs and landings on 30-31 July.





**Figure 10 :** As in Fig. 4, except for typhoon Megi on 26 September 2016 00 UTC, and storm motion (5.5 m s$^{-1}$, 300°).

**Figure 11 :** As in Fig. 5, except for the six takeoffs from and landings to TPE of IAGOS aircraft on 25-26 September.

**Figure 12 :** As in Fig. 6, except for the period 25 September 12 UTC - 26 September 18 UTC, and 6 tracks of IAGOS aircraft during takeoffs and landings on 25-26 September.

**Figure 13 :** A schematical cross-section (not on scale) of a model typhoon and associated transports of ozone and carbon monoxide, in relation with dry and moist regions, and high potential vorticity zones (see text).



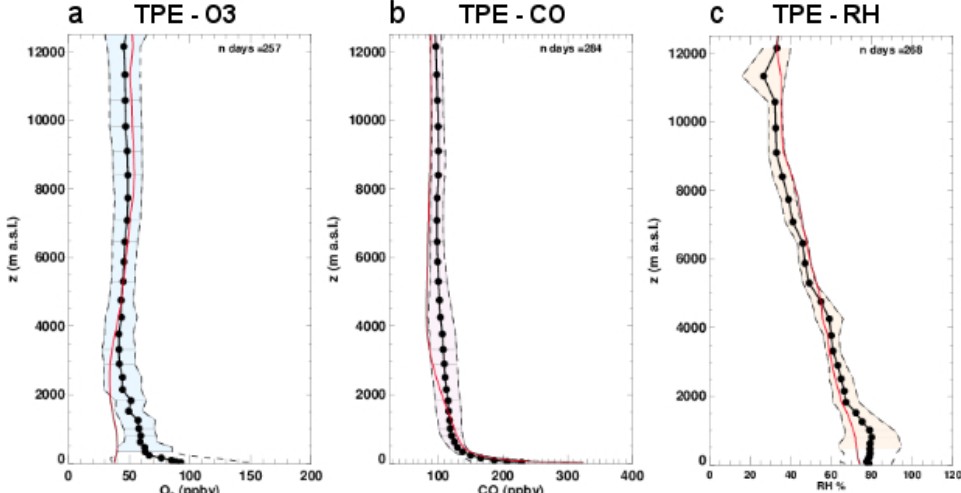

**Figure 1 :** Mean profiles of **(a)** ozone, **(b)** carbon monoxide, **(c)** relative humidity from IAGOS measurement during take–off from and landing to TPE over the time period 1 July to 31 October 2012 to 2015 (in black) along with the standard deviation indicated by the shaded area, and for 2016 (in red). The total number of days with data is indicated.

35





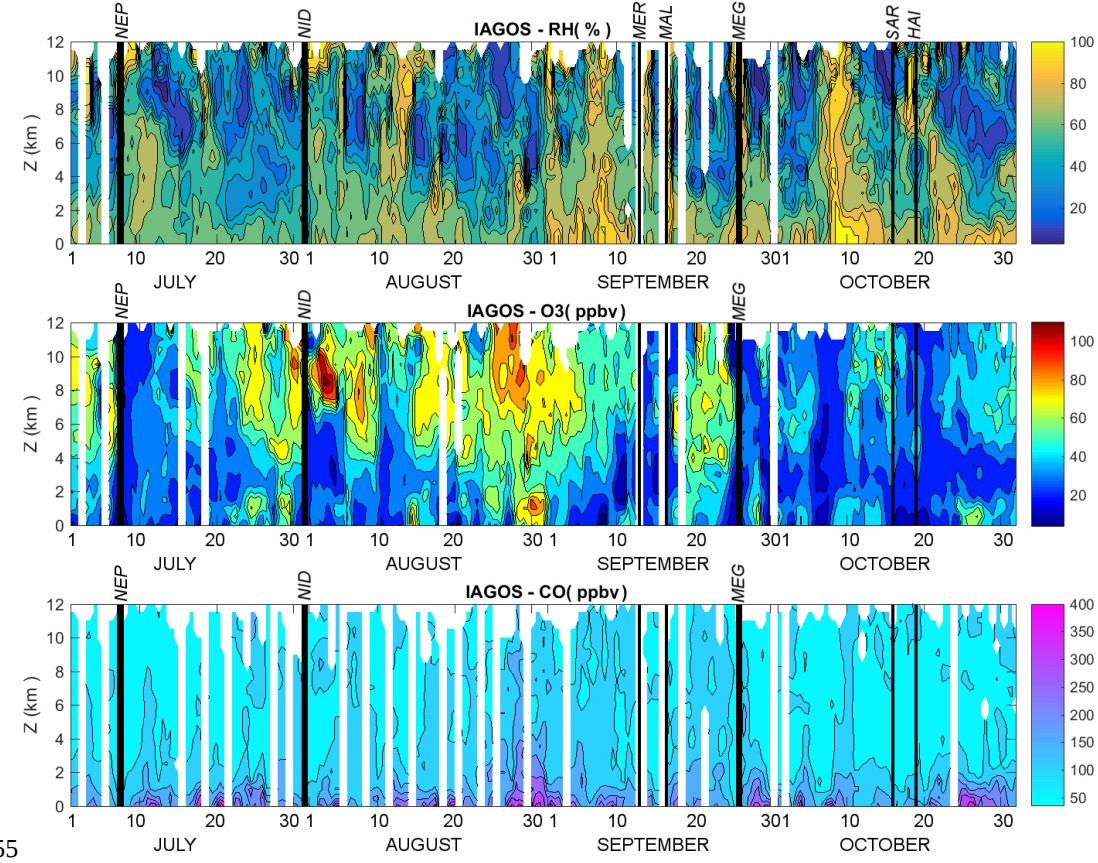

**Figure 2 :** Vertical profiles derived from IAGOS measurements during take-off from and landing to TPE (Taoyuan International Airport ; 25.076°N, 121.224°E ) from 1 July till 31 October 2016 for **(a)** relative humidity in %, **(b)** ozone mixing ratio in parts per billion by volume – ppbv, **(c)** carbon monoxide in ppbv. Vertical black lines labeled NEP, NID, MER, MAL, MEG, SAR, HAI represent the closest approaches of typhoons Nepartak, Nida,

Meranti, Malakas, Megi, Sarika and Haima, respectively.





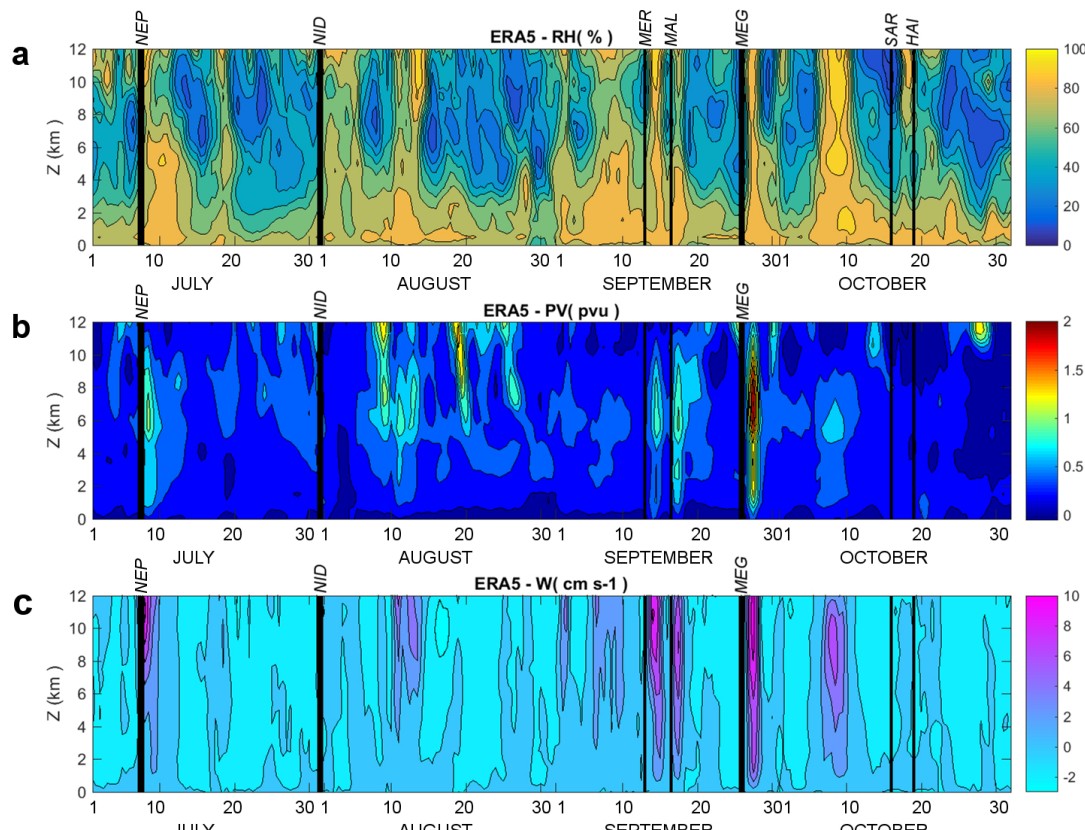

**Figure 3 :** Vertical profiles derived from ERA–5 re–analyses within 300 km from TPE from 1 July till 31 October 2016 for

(a) mean relative humidity in %, (b) mean potential vorticity in potential vorticity unit – 1 PVU = $10^{-6}$ $m^2$ $s^{-1}$ K $kg^{-1}$, (c) mean vertical velocity in cm $s^{-1}$. Vertical black lines labeled NEP, NID, MER, MAL, MEG, SAR, HAI represent the closest approaches of typhoons Nepartak, Nida, Meranti, Malakas, Megi, Sarika and Haima, respectively.





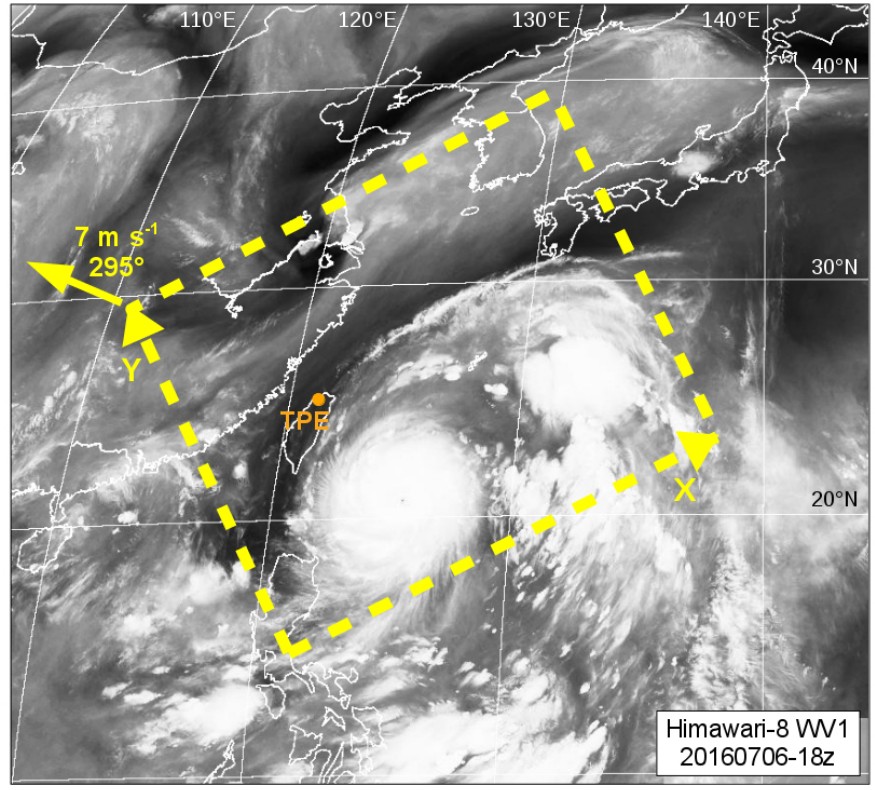


**Figure 4 :** Himawari-8 Water Vapor / Channel 8 (6.06 - 6.43 µm) image on 6 July 2016 18 UTC showing typhoon Nepartak centered about 550 km southwest of Taiwan. Orange dot labelled TPE indicates the location of the Taiwan Taoyuan International airport. The yellow contour denotes the 2000 km × 2000 km region where ERA−5 re−analyses data were composited (X and Y axes are shown). The storm motion (7 m s⁻¹, 295°) is indicated.




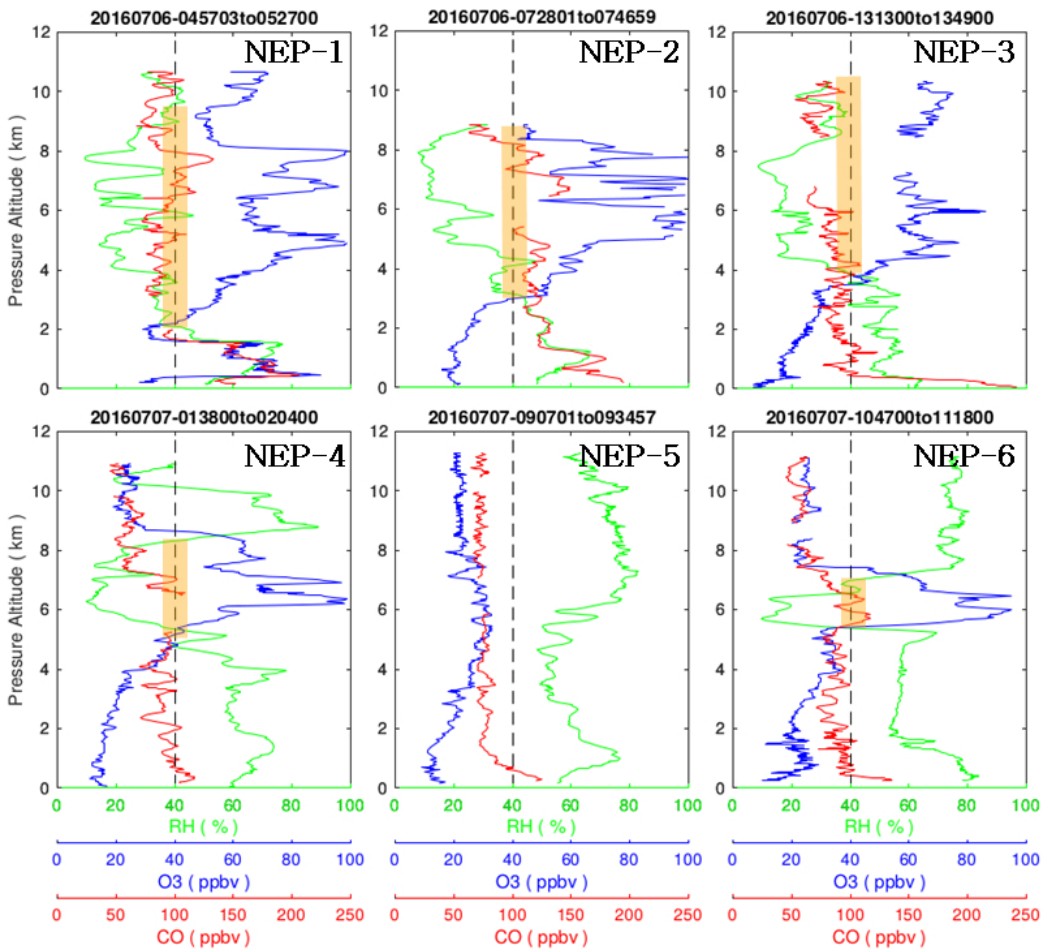

**Figure 5 :** Vertical profiles of relative humidity RH (green curves, in %), ozone O3 (blue curves, in parts per billion by volume – ppbv) and carbon monoxide CO (red curves, in ppbv) from the six takeoffs from and landings at TPE of IAGOS aircraft on 6-7 July, in the vicinity of typhoon Nepartak. The dashed line is an arbitrary limit separating dry (RH <40 %) and moist (RH >40 %), O3-poor (<40 ppbv) and O3-rich (>40 ppbv), CO-poor (<100 ppbv) and CO-rich (>100 ppbv) air masses. Thick orange segments indicate regions with dry and O3-rich air.






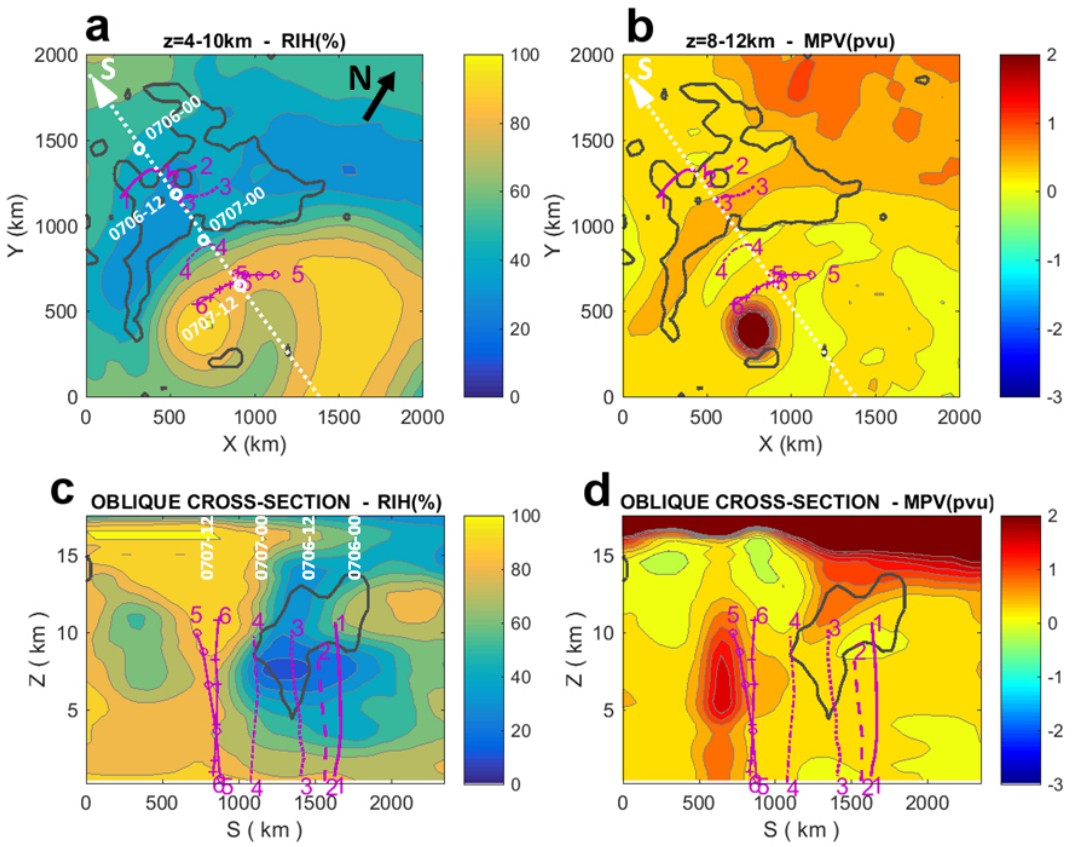

**Figure 6 :** Composites of ERA-5 re-analyses data from 6 July 00 UTC till 7 July 12 UTC in the domain shown in Fig. 4. Above are horizontal distributions of **(a)** relative integrated humidity (RIH) between 4 and 10 km altitude (in %), and **(b)** mean potential vorticity (MPV) between 8 and 12 km altitude in PVU (= $10^{-6}$ m$^2$ s$^{-1}$ K kg$^{-1}$). The north direction is indicated in **(a)**. The dotted white line (S axis) represents the pseudo-trajectory of TPE airport in the frame moving with Typhoon Nepartak (successive times are indicated). Below are vertical cross-sections of **(c)** RIH in %, and **(d)** MPV in PVU, averaged across ± 500 km along this oblique S line. The solid black contours encompass mean downward velocities < -1 cm s$^{-1}$ between 6 and 10 km altitude. The purple lines 1 to 6 indicate the tracks of IAGOS aircraft during selected takeoffs and landings on 6-7 July.

23



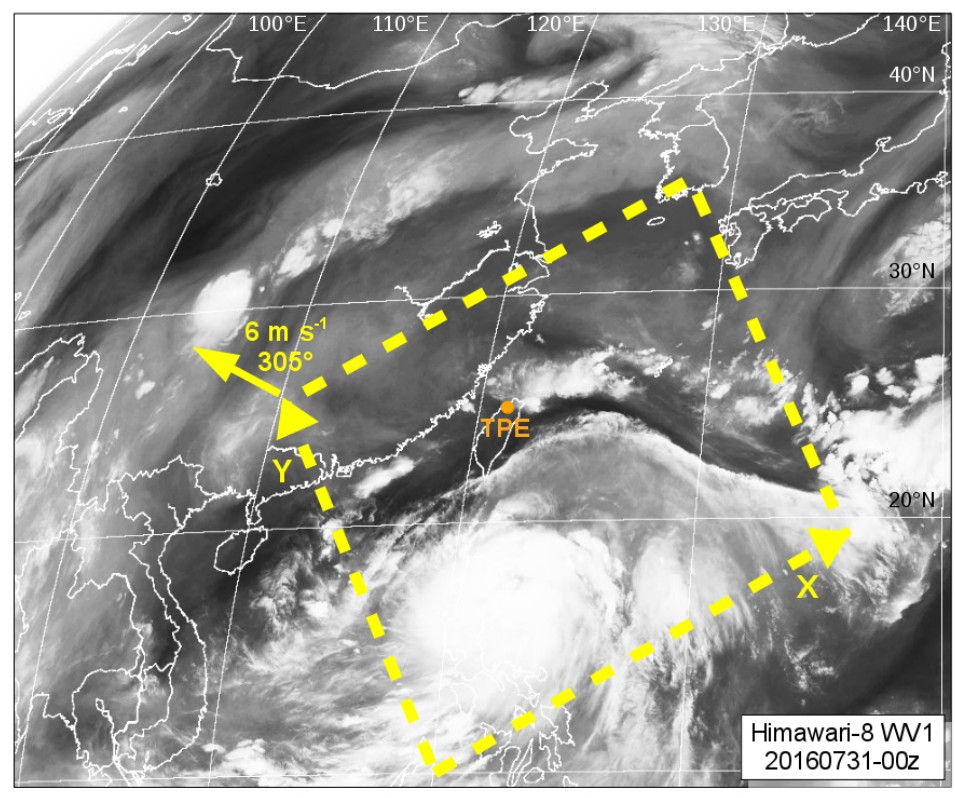


**Figure 7 :** As in Fig. 4, except for severe tropical storm Nida on 31 July 2016 00 UTC, and storm motion (6 m s$^{-1}$, 305°).





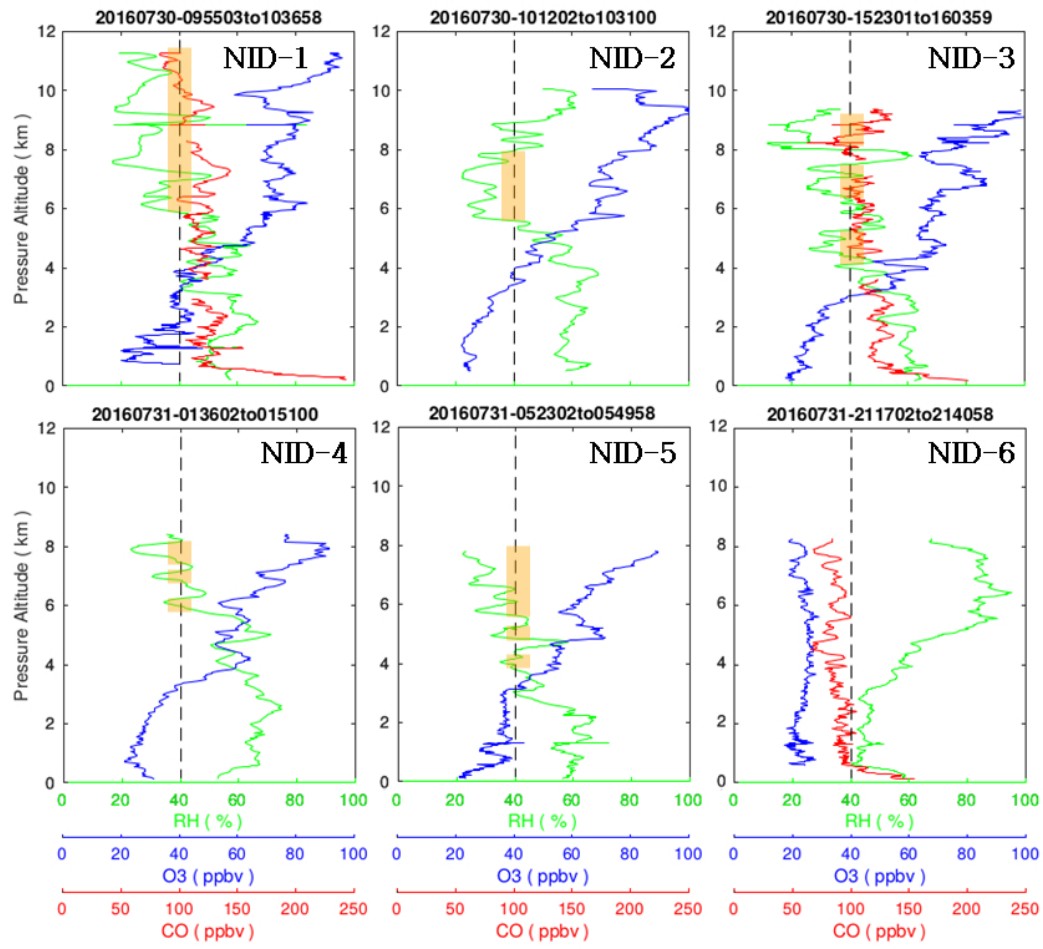

700**Figure 8 :** As in Fig. 5, but for the six takeoffs from and landings at TPE of IAGOS aircraft on 30-31 July in the vicinity of
severe tropical storm Nida.



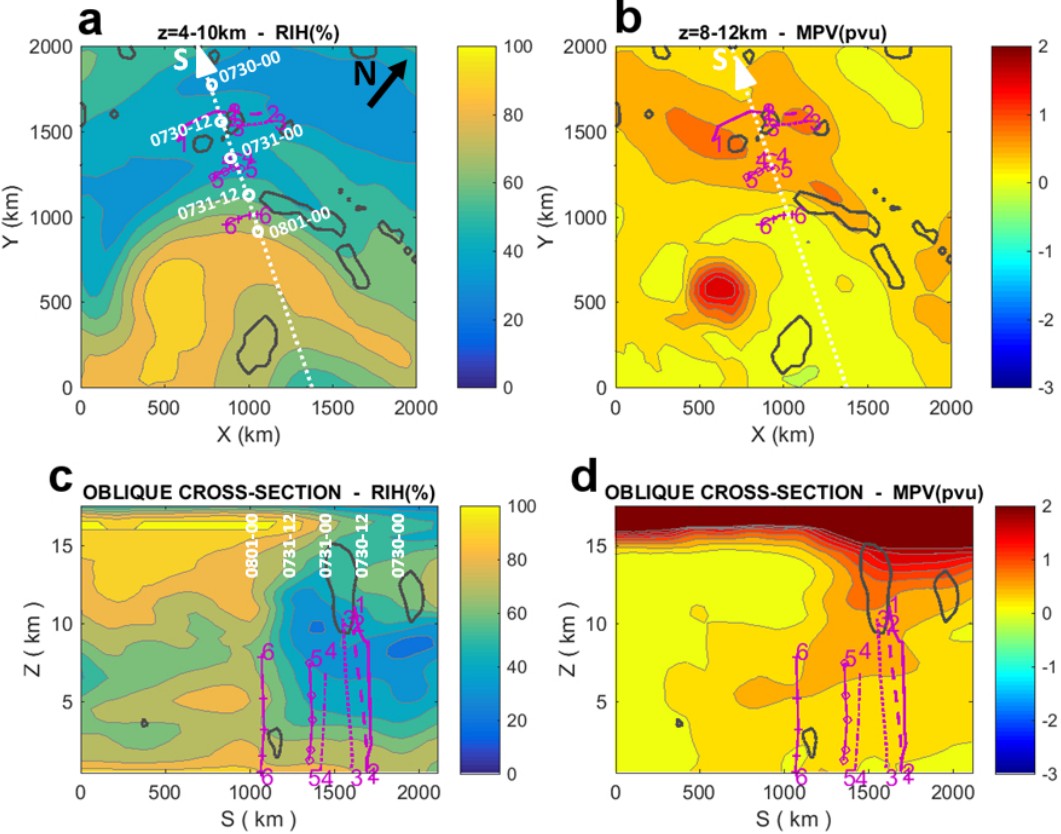

**Figure 9 :** As in Fig. 6, but for the period 30 July 06 UTC - 01 August 00 UTC, and 6 tracks of IAGOS aircraft during
takeoffs and landings on 30-31 July in the vicinity of severe tropical storm Nida.



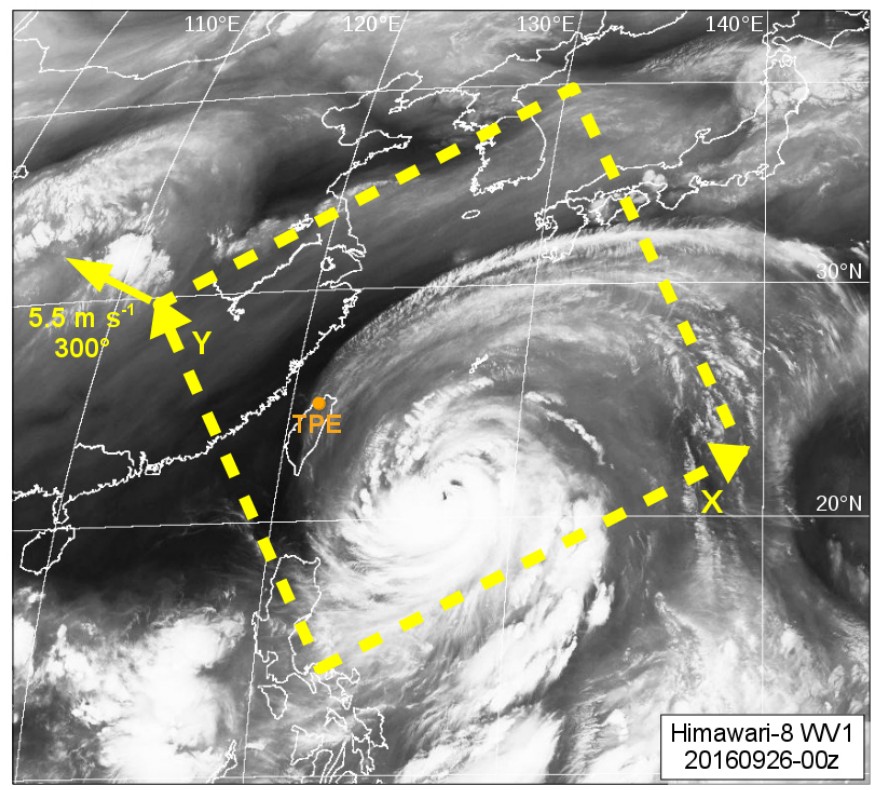

**Figure 10 :** As in Fig. 4, except for typhoon Megi on 26 September 2016 00 UTC, and storm motion (5.5 m s⁻¹, 300°).






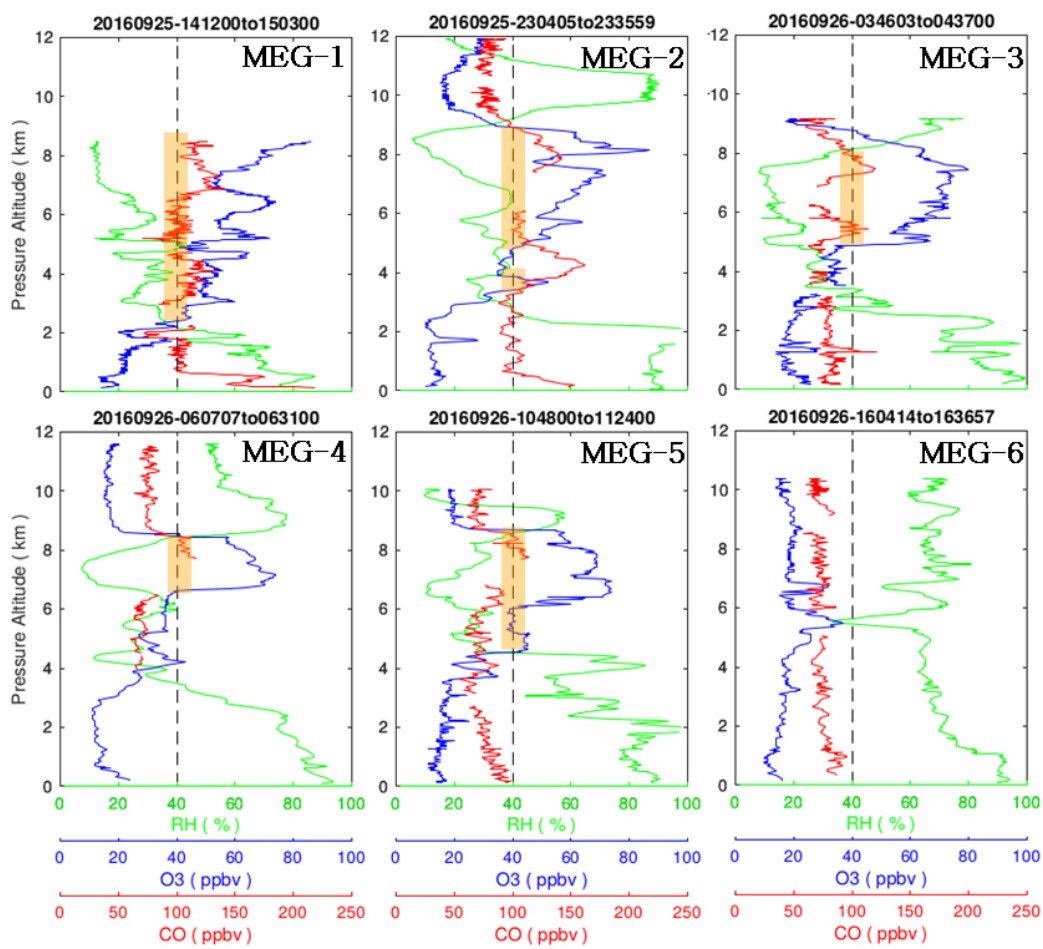

**Figure 11 :** As in Fig. 6, but for the six takeoffs from and landings at TPE of IAGOS aircraft on 25-26 September shown in Fig. 11, in the vicinity of Typhoon Megi.




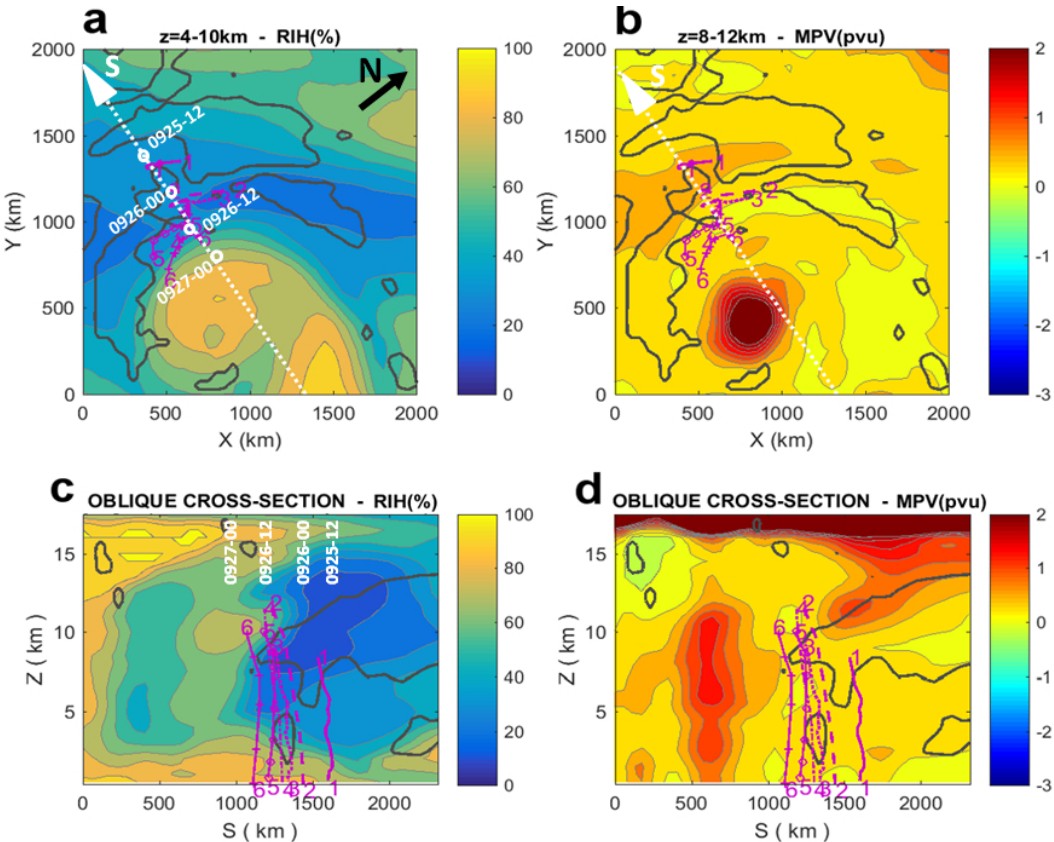

**Figure 12 :** As in Fig. 5, but for the period 25 September 12 UTC - 26 September 18 UTC, and 6 tracks of IAGOS aircraft during takeoffs and landings on 25-26 September, in the vicinity of Typhoon Megi.





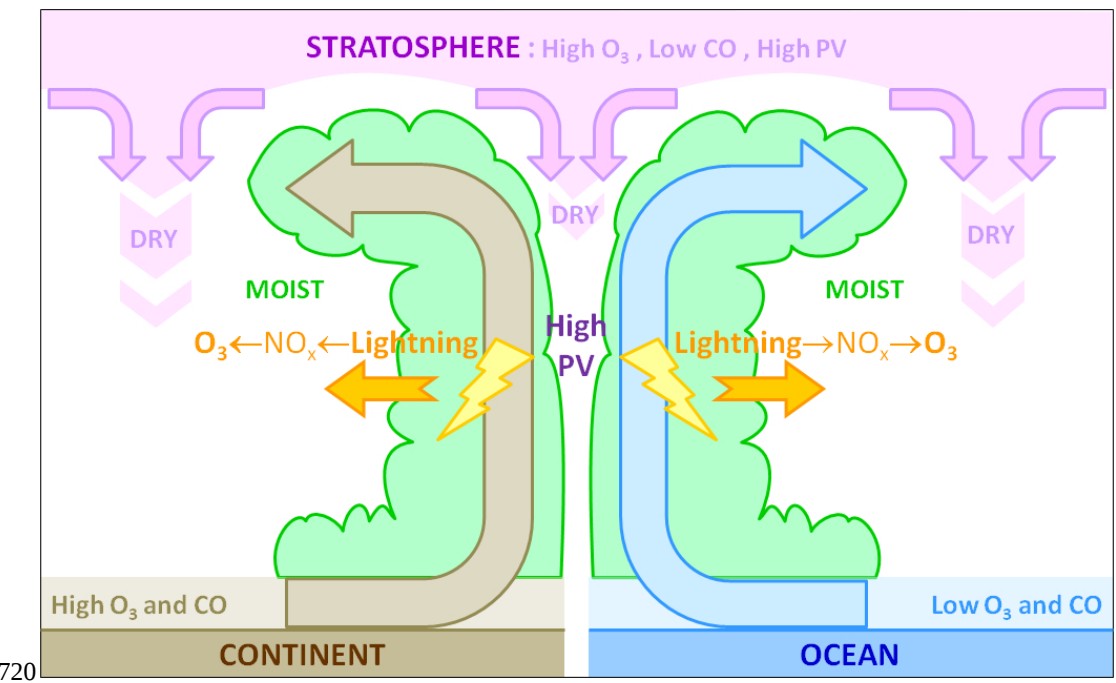


**Figure 13 :** A schematical cross-section (not on scale) of a model typhoon and associated transports of ozone and carbon monoxide, in relation with dry and moist regions, and high potential vorticity zones (see text).




60