# Peer review of "The influence of typhoons on atmospheric composition deduced from IAGOS measurements over Taipei"

_Atmospheric Chemistry and Physics, 2019_

## Referee Comment (RC1) · Anonymous Referee #1 · 23 Oct 2019

Comments on the manuscript entitled, 'The influence of typhoons on atmospheric composition deduced from IAGOS measurements over Taipei', Roux et al. submitted for plausible publication in Atmos. Chem. Phys.

This paper deals with the effects of typhoons on ozone concentration and relative humidity distribution in the upper troposphere over Taipei (Taiwan) using IAGOS data sets collected by two China Airlines during summer 2016. This study is very important, in principle, since detail knowledge of water vapour and ozone budget in the upper tropospheric plays a vital role in a global weather-climate system. The paper is well written and contains significant data and original materials. However, the manuscript needs

moderate revision before publications. My specific comments are following under :

(1) The authors have shown a significant increase in ozone concentration in the proxy of dry air and low CO concentration which occurs in the middle and upper troposphere. I do agree with the authors that the enhanced ozone is primarily associated with the passage of typhoon. But my main concern, even the authors themselves have stated that lightning associated with a typhoon may cause enhancement of ozone. Then how sure we are that the observed enhanced ozone in the middle and upper troposphere is of stratospheric origin. I suggest doing a back-trajectory analysis. (2) Potential vorticity (PV): PV is a dynamical tracer for stratosphere-troposphere exchange (STE) processes but in the absence of diabatic heating and frictional forces. During a typhoon, the convective tower will have a high value of PV, which is generated due to latent heating associated with it. A detailed study on the evolution of PV structures associated with a typhoon has been carried out by Grad et al. (2011). Thus one needs to take caution during a typhoon while linking high PV value as the air of stratospheric origin. However, Leclair De Bellevue et al. (2007) have mentioned that latent heat can be negligible outside the convection core in the upper troposphere. In this aspect, a detailed discussion is required in the manuscript. (3) There may be a thermal sensor in the aircraft. Thus it will be good to show the thermal structure (maybe temperature inversion). (4) Near Taipei, there is an ozonesonde launching station at Banqiao (25 degree N, 121.3 degree E) or elsewhere. It will be quite supportive of the IAGOS observations if one can show the ozonesonde profiles during any of the typhoon cases (till 3-4 days after typhoon landfall). This will also validate IAGOS data in the convective situation. (5) The tropopause structure has a key role in STE and also during a typhoon. Thus, I suggest authors take nearby radiosonde data (twice a day) to show the thermal structure and also the wind information. (6) Ratnam et al. (2016) have shown a significant increase in upper tropospheric ozone associated with north Indian Ocean tropical cyclones. It is shown that a particular sector of the cyclone has high ozone and low humidity. I also suggest the authors look into this possibility in the existing data set and discuss the results. (7) Figs.4, 7 and 10 can be combined.

References :

Graf, M.A., Sprenger, M., Moore, R.W., 2011. Central European tornado environments as viewed from a potential vorticity and Lagrangian perspective. Atmos. Res. 101, 31–45. oi:10.1016/j.atmosres.2011.01.007.

Leclair De Bellevue, et al. 2007. Simulations of stratospheric to tropospheric transport during the tropical cyclone Marlene event. Atmos. Environ. 41, 6510–6526.

Ratnam M.V., et al. 2016. Effect of tropical cyclones on the stratosphere–troposphere exchange observed using satellite observations over north Indian Ocean. Atmos Chem Phys 16:8581–8591. https ://doi. org/10.5194/acp-16-8581-2016

---

## Referee Comment (RC2) · Laura Pan (Referee) · 19 Dec 2019

The manuscript by Roux et al. presents an excellent data analysis, revealing the chemical transport structure of typhoons over the Pacific using IAGOS data. As described in the manuscript, the discussion section in particular, stratosphere-to-troposphere transport induced by convective storms has been observed and analyzed in a number of previous studies. This study, however, provides the clearest 3D structure of the process. The technique of analysis, translating the time evolution of the typhoon systems into spatial extent using takeoff and landing profiles from/to a single airport (Taipei/Taoyuan), is a highly effective way to use the data. Integrating new generation

of re-analysis ERA5 and satellite maps of the storms, the chemical profiles are nicely connected to the anatomical structure of the typhoon system.

Specifically, I find it a very good diagnostic to identify the consistency between the O3-RH relationship from measurements and the PV-RH relationship from the ERA5. This pair of "tracers" allows the author to separate the convection generated high PV in the center of the storms from the stratospheric influenced high PV air wrapped at the outer rim of the typhoon circulation.

Overall, I find the result of this work important and the method of integrating meteorological information and chemical measurements inspiring. I recommend the manuscript to be accepted for publication largely as it is, with some minor corrections. I have some suggestions, optional for the authors, which may enhance the take home message of the work.

Minor changes and corrections:

1. Page 4: line 142: "where F is.." => No F in the equation

2. Page 7, Line 267: this sentence needs to be revised. The word "determine" is too strong. "...two more cases to highlight the common features"?

3. Page 8, you made a number of references of "low CO" to values of ∼100 ppbv, which is somewhat problematic. Overall, the CO signatures are weak.

4. Figures:

\* Figure 1 labels are too small to read on a print page.

\* Fig 4: "Yellow box"

Additional suggestions:

1) Figure 1 serves a purpose but could have much more information content. As shown in later profiles, the individual profiles could be very structured. These mean profiles,

however, are kind of uninteresting. Suggest you to try "box-and-whisker" plots with distribution in layers instead of mean and standard deviation.

2) Figure 2&3 could use some color adjustment to highlight the features you want readers to see. Panels C in particular. Fig. 2C could be clearer if color changes for CO above/below 100 ppb and 50 ppbv. Fig. 3C should show color change at 0 to highlight direction change

3) After presenting the details of three cases, it would be more satisfying to have a summary figure quantifying the layer influenced by the stratospheric air, its vertical extent and the amount of ozone enhancement. It is possible to do this using all 18 profiles but present the data in the tracer-tracer space. For an example, see Fig. 3 of Randel et al., 2016. If you have a large anti-correlation between O3 and RH, it would be a strong support for transport. The part of the tracer space with positive ozone anomaly and unclear anti-correlation with RH may indicate other mechanisms, including lightening NOx facilitated ozone production.

Randel, W.J., L. Rivoire, L. L. Pan and S. B. Honomichl (2016), Dry layers in the tropical troposphere observed during CONTRAST and global behavior from GFS analyses, J. Geophys. Res. Atmos., 121, 14,142–14,158, doi:10.1002/2016JD025841.

---

## Author Comment (AC1) · 28 Jan 2020

Anonymous Referee #1 Comments on the manuscript

(...) the manuscript needs moderate revision before publications. My specific comments are following under :

(1) The authors have shown a significant increase in ozone concentration in the proxy of dry air and low CO concentration which occurs in the middle and upper troposphere. I do agree with the authors that the enhanced ozone is primarily associated with the passage of typhoon. But my main concern, even the authors themselves have stated

that lightning associated with a typhoon may cause enhancement of ozone. Then how sure we are that the observed enhanced ozone in the middle and upper troposphere is of stratospheric origin. I suggest doing a back-trajectory analysis.

-> Retrotrajectory analyses have been added for typhoons Nepartak, Nida and Megi. FLEXPART Lagrangian dispersion model( Stohl et al. 2005) has been used in order to determine the origin of ozone-rich layers identified with IAGOS pseudo-vertical soundings. As discussed in the revised version, the backward plumes of particles initiated within these layers reveal a probable stratospheric origin 2.5 to 4 days before IAGOS observations. Associated Figs. 8 (for Nepartak), 11 (for Nida) and 14 (for Megi) have been added in the revised version.

(2) Potential vorticity (PV): PV is a dynamical tracer for stratosphere-troposphere exchange (STE) processes but in the absence of diabatic heating and frictional forces. During a typhoon, the convective tower will have a high value of PV, which is generated due to latent heating associated with it. A detailed study on the evolution of PV structures associated with a typhoon has been carried out by Grad et al. (2011). Thus one needs to take caution during a typhoon while linking high PV value as the air of stratospheric origin. However, Leclair De Bellevue et al. (2007) have mentioned that latent heat can be negligible outside the convection core in the upper troposphere. In this aspect, a detailed discussion is required in the manuscript.

-> A sentence has been added at the end of 2.3. The use of potential vorticityÂǎ: In the upper troposphere outflow layer, at large distances ($\geq$500 km) from the cyclone core, where diabatic heating and friction are small, PV must be nearly conserved following air parcels and may be a good indicator of large-scale horizontal and vertical motions (e.g. Molinari et al., 1998 ; Leclair de Bellevue et al., 2007).

(3) There may be a thermal sensor in the aircraft. Thus it will be good to show the thermal structure (maybe temperature inversion).

-> We have compared the vertical profiles of relative humidity, ozone and carbon monoxide contents (previous Figs. 5, 8 and 11), with the vertical profiles of pressure, temperature T and potential temperature $\theta$ (see the attached Profiles_RHO3CO+PTTH_Nep+Nid+Meg.pdf). No systematic association could objectively be found between the base and the top of the dry and ozone-rich layers (orange dashed lines), and the vertical gradients of T or $\theta$. Nevertheless, small changes in dT/dz and d$\theta$/dz can be seen for profiles NEP-1 at 2.2 km altitude, NEP-2 at 3 km, NEP-6 at 5.4 and 7.2 km, NID-5 at 3.2 km, MEG-2 at 9 km, MEG-4 at 6.4 and 8,4 km, MEG-5 at 4.5 km. Considering this meager evidence, this discussion was not included in the revised version.

(4) Near Taipei, there is an ozonesonde launching station at Banqiao (25 degree N, 121.3 degree E) or elsewhere. It will be quite supportive of the IAGOS observations if one can show the ozonesonde profiles during any of the typhoon cases (till 3-4 days after typhoon landfall). This will also validate IAGOS data in the convective situation.

-> Unfortunately, the Central Weather Bureau did not conduct ozonesonde measurements in June-October 2016.

(5) The tropopause structure has a key role in STE and also during a typhoon. Thus, I suggest authors take nearby radiosonde data (twice a day) to show the thermal structure and also the wind information.

-> Indeed, these radiosounding data reveal that the "cold point tropopause" decreased substantially when the dry zone arrived over Taiwan (Typhoon NepartakÂǎ: from 18.2 km on 5 July 00 UTC to 16.2 km on 6 July 00 UTC, Typhoon Megi Âǎ: from 17.3 km on 25 September 00 UTC to 16.1 kmon 26 September 00 UTC, no significant change was found for Tropical Storm Nida). These remarks have been included in the revised version.

(6) Ratnam et al. (2016) have shown a significant increase in upper tropospheric ozone associated with north Indian Ocean tropical cyclones. It is shown that a particular sector of the cyclone has high ozone and low humidity. I also suggest the authors look

into this possibility in the existing data set and discuss the results.

-> A reference to Venkam Ratman et al. (2016) has been added in the revised version.

(7) Figs.4, 7 and 10 can be combined.

-> Figs. 4, 7 and 10 have been combined into Fig. 5 in the revised version.

We thank Referee #1 for the constructive comments which helped to clarify the submitted version.

Please also note the supplement to this comment:
https://www.atmos-chem-phys-discuss.net/acp-2019-622/acp-2019-622-AC1-supplement.pdf

**Supplement:**

**NEP-1**

**NEP-2**

**NEP-3**

[Figure]

**NEP-4**

**NEP-5**

**NEP-6**

[Figure]

[Figure]

**NID-1**

[Figure]

**NID-2**

[Figure]

**NID-3**

[Figure]

**NID-4**

[Figure]

**NID-5**

[Figure]

**NID-6**

[Figure]

**MEG-1**

[Figure]

**MEG-2**

[Figure]

**MEG-3**

[Figure]

**MEG-4**

[Figure]

**MEG-5**

[Figure]

**MEG-6**

---

## Author Comment (AC2) · 28 Jan 2020

(...) I recommend the manuscript to be accepted for publication largely as it is, with some minor corrections. I have some suggestions, optional for the authors, which may enhance the take home message of the work.

Minor changes and corrections:

1. Page 4: line 142: "where F is.." => No F in the equation

-> Equation (2) has been correctedÂă:

2. Page 7, Line 267: this sentence needs to be revised. The word "determine" is too

strong. ". . .two more cases to highlight the common features"?

-> "determine" has been changed into "investigate" in the revised version.

3. Page 8, you made a number of references of "low CO" to values of âĹij100 ppbv, which is somewhat problematic. Overall, the CO signatures are weak.

-> "low CO values" have been replaced by "CO values less than 100 ppbv" in the revised version.

4. Figures: 4.1. Figure 1 labels are too small to read on a print page. -> Figure 1 has been changed (see below) in the revised version.

4.2. Fig 4: "Yellow box" -> The change has been made in the revised version.

Additional suggestions:

1) Figure 1 serves a purpose but could have much more information content. As shown in later profiles, the individual profiles could be very structured. These mean profiles,however, are kind of uninteresting. Suggest you to try "box-and-whisker" plots with distribution in layers instead of mean and standard deviation.

-> In the revised version, Fig. 1 shows "boxplots" of RH, O3 and CO for each 500-m deep layer between the surface and 12 km altitude. Comments have been changed accordingly.

2) Figure 2&3 could use some color adjustment to highlight the features you want readers to see. Panels C in particular. Fig. 2C could be clearer if color changes for CO above/below 100 ppb and 50 ppbv. Fig. 3C should show color change at 0 to highlight direction change

-> The color codes for Figs. 2c (IAGOS CO), 3b (ERA-5 PV) and 3c (ERA-5 W) has been changed in the revised version in order to highlight contrasts.

3) After presenting the details of three cases, it would be more satisfying to have a

summary figure quantifying the layer influenced by the stratospheric air, its vertical extent and the amount of ozone enhancement. It is possible to do this using all 18 profiles but present the data in the tracer-tracer space. For an example, see Fig. 3 of Randel et al., 2016. If you have a large anti-correlation between O3 and RH, it would be a strong support for transport. The part of the tracer space with positive ozone anomaly and unclear anti-correlation with RH may indicate other mechanisms, including lightening NOx facilitated ozone production.

-> In the revised version, a figure (Fig. 4) has been added to show the two-dimensional distribution of O3 and RH from 4 to 12 km altitudes for 56 vertical profiles derived from IAGOS measurements during take-off from and landing to TPE at less than 1200 km from the centers of 7 storms that passed relatively close to TPE during July to October 2016. As discussed in the revised version, two populations can be identified : ozone-rich (50 to 100 ppbv) and dry (RH < 50%) air represents 35.8 % of the measurements, ozone-poor (<50 ppbv) air with various RH values (20 to 100 %) accounts for the remaining 64.2 %. The absence of measurements with positive ozone anomaly and high relative humidity indicates that lightning-induced nitrogen oxides are probably not the main cause of ozone production here. These results bear some resemblance with those Randel et al. (2016) derived from vertical tropospheric profiles collected during the "Convective Transport and Active Specied in the Tropics" experiment from Guam (14°N, 145°E) in January-February 2014. They also observed dry air (RH < 20%) with enhanced ozone (40-80 ppbv) contents within layers between 3 and 9 km altitudes, which they linked to quasi-isentropic transport from the extratropical UTLS. These data were obtained in the descending branch of the Hadley cell, with only occasional deep convection, which contrasts with the moister and stormier conditions that prevailed during July to October 2016 near Taiwan.

We thank Laura Pan for the constructive comments which helped to clarify the submitted version.

[Figure]

2019.